# Regulatory T Cells with Additional COX-2 Expression Are Independent Negative Prognosticators for Vulvar Cancer Patients

**DOI:** 10.3390/ijms23094662

**Published:** 2022-04-22

**Authors:** Nadine Ansorge, Christian Dannecker, Udo Jeschke, Elisa Schmoeckel, Helene Hildegard Heidegger, Aurelia Vattai, Maximiliane Burgmann, Bastian Czogalla, Sven Mahner, Sophie Fuerst

**Affiliations:** 1Department of Obstetrics and Gynecology, University Hospital, LMU Munich, Marchioninistrasse 15, 81337 Munich, Germany; nadine.ansorge@uk-augsburg.de (N.A.); helene.heidegger@med.uni-muenchen.de (H.H.H.); aurelia.vattai@med.uni-muenchen.de (A.V.); maximiliane.burgmann@med.uni-muenchen.de (M.B.); bastian.czogalla@med.uni-muenchen.de (B.C.); sven.mahner@med.uni-muenchen.de (S.M.); sophie.fuerst@med.uni-muenchen.de (S.F.); 2Department of Obstetrics and Gynecology, University Hospital Augsburg, Stenglinstrasse 2, 86156 Augsburg, Germany; christian.dannecker@med.uni-augsburg.de; 3Department of Pathology, LMU Munich, Thalkirchner Str. 36, 80337 Munich, Germany; elisa.schmoeckel@med.uni-muenchen.de

**Keywords:** vulvar cancer, regulatory T cells, COX-2, tumor-infiltrating lymphocytes, M2-polarized macrophages

## Abstract

Vulvar cancer incidence numbers have been steadily rising over the past decades. In particular, the number of young patients with vulvar cancer has recently increased. Therefore, the need to identify new prognostic factors and, in addition, therapeutic options for vulvar carcinoma is more apparent. The aim of this study was to analyze the influx of COX-2 positive tumor-infiltrating lymphocytes and monocytes and their influence on prognosis. Using subtyping by immunofluorescence, the majority of COX-2 expressing immune cells were identified as FOXP3-positive regulatory T cells. In addition, peri- and intra-tumoral macrophages in the same tumor tissue were detected simultaneously as M2-polarized macrophages. COX-2 positive immune cells were independent negative prognostic markers in long-term overall survival of patients with vulvar cancer. These results show an influence of immune cell infiltration for vulvar carcinoma patients. Immune cell infiltration and immune checkpoint expression may, therefore, become interesting targets for further research on new vulvar cancer treatment strategies.

## 1. Introduction

In 2020, more than 17,000 women worldwide died from vulvar cancer. The number of new cases has been steadily increasing over recent decades, with a worldwide incidence of 45,240 new cases [1]. An additional threat is the fact that a continuous increase in new cases in young women has been observed [2,3]. A total of 90% of vulvar carcinomas are squamous cell carcinomas (VSCC). Non-keratinized squamous cell carcinomas are often human papilloma virus (HPV)-associated and mainly affect younger women [4,5]. Essential for malignant transformation in HPV-dependent carcinogenesis is the inactivation of p53 and the retinoblastoma tumor-suppressor gene product by the viral gene products E6 and E7 [6]. In contrast, keratinized squamous cell carcinomas are usually HPV-independent and, due to chronic genital inflammatory disease, such as lichen sclerosus, affect older women [7,8]. Beside HPV, the development of vulvar carcinomas is associated with other risk factors: immunosuppression, smoking [9], and sexually transmitted diseases, such as herpes simplex virus 2 infections [5], are associated with an increased risk of this disease. Radical surgical interventions are predominantly used for therapy, often ending in vulvectomy in the case of extensive involvement. To date, the mental consequences of such a serious and extensive intervention have not been studied extensively. Sexual behavior restrictions, micturition problems, or even mental effects that impair quality of life are long-term consequences of this radical form of therapy [10,11]. As for prevention, HPV vaccination, for example, has been seen as a new hope in fighting against HPV-related tumors such as cervical, anal, and vulvar cancers [12,13]. In 2016, the EURO Vaccination Meeting listed Belgium as the top country with a vaccination rate of 84%, while in Germany the vaccination rate reached a critical 31% in 2015 [14]. Based on the low vaccination rate and an aging population in Germany it can be assumed that the need for newly found prognostic factors for vulvar carcinoma is even more apparent. Our group showed negative prognosticators in vulvar cancer patients, such as LDOC1 [15] or combined expression of COX-2 and PPARγ in cytoplasm of vulvar cancer tissue [16]. An increasing number of new cases, younger patients, radical therapy, and a lack of comprehensive prevention due to the low vaccination rates, at least in this country, are concerning observations. Cyclooxygenase-2 (COX-2) features as a long-standing object of scientific interest in the context of carcinogenesis in several tumor entities [17,18,19,20,21]. COX-2, in contrast to the constitutive housekeeping enzyme COX-1, is inductively expressed as a known inflammatory enzyme [22,23]. Tissues of the brain, kidney, testis, and tracheal epithelium are exceptions with respect to constitutive COX-2 expression [24,25]. The inducing of the COX-2 enzyme is triggered by cell damage or inflammation by the release of various factors, such as growth factors like epidermal growth factor (EGF) [26], prostaglandins, or chemokines like TNF-γ [27]. Affecting COX-2 products, the prostanoids appear to be associated with the development and progression of tumor disease. Factors such as angiogenesis, invasion, apoptosis inhibition, growth, and aggressiveness of the tumor seem to be highly dependent on COX-2 and its products [28,29]. It is thought that products of COX-2, such as prostaglandin E2 (PGE2), critically influence the development of tumors e.g., in angiogenesis [30,31]. COX-2 is also known to be active in cancer-associated immune cells [32,33]. We used the abbreviation sTILs (stromal tumor-infiltrating lymphocytes) in our study for these cells. They have become important players in immuno-oncology with regard to predicting prognosis in cancer patients [34,35,36,37]. Already in the treatment of breast carcinomas, the presence of sTILs is taken into account in the interpretation of tumor biology and ultimately in the decision-making process of the optimal therapy according to guidelines [38]. Due to the diverse cell populations within sTILs, the subgroups are not fully understood in their prognostic role and are potentially conceivable as biomarkers in the future [39]. 

Types of sTILs and iTILs form a specific group of immune cells called tumor-infiltrating lymphocytes (TILs) and were assessed based on the recommendations of the International TIL Working Group (ITILWG). According to this working group, lymphocytes organized in tumor nests are defined as intratumoral TILs (iTILs). They making cell-to-cell contact without intervening stroma or interacting directly with carcinoma cells. Hence, stromal TILs (sTILs) are located in the stroma among carcinoma cells and have no direct contact with carcinoma cells [40]. 

In this study, we specifically investigated sTILs and no iTILs. Our study analyzed and characterized sTILs expressing COX-2 as mainly Treg cells in tissues of VSCC, their relevance as a prognostic factor, and in addition, the subtyping and polarization of infiltrating macrophages in the tumor microenvironment.

## 2. Results

### 2.1. High COX-2 Intensity of sTILs as an Independent Negative Prognostic Factor in Long-Term Overall Survival

Figure 1A,B show the sTILs with low and high intensity of COX-2. The Kaplan–Meier curve shows a significant survival disadvantage for patients whose sTILs have a COX-2 intensity > 2 in overall survival, especially in long-term survival from 60 months (Figure 1C). A total of 47% of the tumor samples have a COX-2 intensity > 2 for COX-2 in sTILs; the remaining 41% were below this intensity value. As the Kaplan–Meier test illustrates, the 10-year overall survival rate of patients with an IRS value > 2 was 52%, but patients with a lower COX-2 intensity lived longer at 79% (Figure 1C). These data show a median survival advantage for patients with lower intensity values (≤2) compared with patients with higher intensity values (>2) at 87 months (Table 1). Multivariate analysis revealed that COX-2 expression in the sTILs of vulvar cancer patients who were alive at 60 months acted as an independent prognostic factor for overall survival (* *p* = 0.007, Table 2). However, tumor stage, nodal status, grading, p16 status and FIGO classification did not act as independent prognostic factors (Table 2).

In addition, the calculations demonstrated for this patient collective that the intensity of COX-2 expression of sTILs has positive correlations with the general percentage of COX-2 expression in tumor tissue (* *p* = 0.001), the IRS score of COX-2 expression in tumor tissue (* *p* = 0.014), and the combined cytoplasmic expression of COX-2 and PPARγ (* *p* = 0.011). An analysis of total COX-2 high stroma cell expression was also performed, although without significant differences. The results are presented as Appendix A. In addition, we also analyzed the influence of tumor COX-2 and sTILs infiltration and found that COX-2 is a negative prognosticator in cases with high stromal COX-2 intensity (Appendix A). 

### 2.2. Significant Majority of COX-2 Positive sTILs Are FOXP3 Positive Treg Cells

The immune cell subpopulations were quantified by counting CD56-, CD68-, and FOXP3-positive cells per field of view (20× magnification). Subtyping COX-2 expressing sTILs by immunofluorescence staining revealed parallel expression of FOXP3 in a clear majority (Figure 2 and Figure 3). 

Because the transcription factor FOXP3 is considered a specific marker of natural CD4 + CD25 + Treg cells, the FOXP3 + COX-2 + sTILs are scored as stromal regulatory T- cells. 70.2% of all counted COX-2 positive sTILs showed concomitant expression of FOXP3 and thus could be detected as Treg cells. There were also in 20.3% CD56 positive sTILs, specific marker for NK cells, and in 9.4% CD68 positive sTILs, macrophages, detected in the subtyping. However, the proportion of these two subtypes was shown to be much lower compared with the proportion of FOXP3 positive sTILs. (** *p* < 0.001, Figure 3).

### 2.3. M2-Polarized Macrophages Are Mainly Located on and in Tumor Tissue

In a further step, the CD68-positive macrophages were differentiated into M2-polarized and non-M2-polarized macrophages using PPARγ as specific marker. In fact, it was already reported that PPARγ plays an essential role as a nuclear receptor for the maturation of alternatively activated M2 macrophages [41] and also primes monocytes into M2 macrophages [42].

M2-polarized macrophages were found to be located peri- to intratumorally and those without M2-polarization remained in the stroma. (Figure 4A,C).

## 3. Discussion

Our study observed COX-2 positive Treg cells to be an independent negative prognosticator in long-term overall survival for vulvar cancer patients (Table 2); also detected in this context are M2-polarized macrophages, which have been previously described as negative influencing factors in several tumor disease [43,44,45].

Regulatory T cells, also known as Treg cells, are also increasingly becoming the focus of research in the field of tumor immunology. As a subset of immunosuppressive T cells, they account for approximately 4–8% of CD4+ T cells in peripheral blood and are characterized by the transcription factor FOXP3 as a specific intracellular marker. Sakaguchi et al. discovered the function of Treg cells as a key role in human immune self-tolerance: a depletion of this cell population in mice showed an induction of autoimmune disease due to their inhibitory effect on CD8+ cytotoxic T cells. [46,47] As already demonstrated in several tumor entities, the expression of Treg cells is found to be a determinant of survival and prognosis of affected patients—but still without definitive knowledge of the pathways in which Treg cells are involved.

Immunological processes in the context of tumorgenesis and maintenance of tumor growth rely on enhancing effect as tumor promotion and on inhibitory effects as tumor suppression. For M2-polarized macrophages and Treg cells in the tumor environment, a tumor-promoting effect was reported [48].

Other studies observed that population of Treg cells in sTILs is significantly higher in carcinoma tissue than in normal healthy tissue [49,50]. Our study shows a clear majority of FOXP3-positive Treg cells with 70.2% (Figure 2 and Figure 3) within the sTILs, so a negative impact on long-term survival in patients with VSCC is assumed. A significant impact of immunologic processes on long-term survival is already known and has been described before [51], but for ovarian cancer Yuan et al. [52] showed that an increased number of Treg cells occur in the tumor microenvironment in patients with gastric carcinoma. There is also an elevated expression of FOXP3 in tumor-infiltrating Treg cells which correlates with up-regulation of COX-2 and its product PGE2 [52]. The COX-2 positivity of the Treg cells can be explained in terms of the COX-2/PGE2 pathway (Figure 5). As Baratelli et al. have established, there is an upregulation of Treg cells by PGE2, an important product of COX-2 [53]. COX-2 as a significant negative prognostic factor on overall survival of the studied patient population has already been demonstrated in our previous study [16]. Studies by Rothwell et al. using COX-inhibitors showed impressive results in terms of improving the prognosis and reducing the incidence of colorectal cancer [54].

Besides Treg cells, we also identified tumor infiltrating macrophages with COX-2 expression. Macrophages are an important component of the innate immune system and are involved in many immunological processes [55]. In cancer research, these immune cells are gaining increasing attention. Tumor-associated macrophages, so-called TAMs, resemble macrophages in regenerating and growing tissue [56]. The different polarization of macrophages, and thereby functional distinction into M1-polarized, tumor-inhibiting macrophages, and M2-polarized, tumor-promoting macrophages, shows the controversial role of this cell group in immunology. Subtyping is indicated by different stimuli. M2-polarization is indicated by Th2 cytokines such as IL-4 and IL-13 and leads to macrophage functionalization in the context of anti-inflammatory processes, tissue repair, and immunoregulatory and tumor-promoting processes [57]. Martinez et al. showed that M2-polarized macrophages exert a major influence on lipid metabolism; this includes induction of COX-2 activity and thus increased production of its enzyme products, such as prostaglandins [58]. For cervical cancer, our group could identify TAMs as a major source of CCL22, a chemokine responsible for Treg cell infiltration [59] (Figure 5).

One of the limitations of our study is the retrospective design. Only a highly limited number of patients included in this collective was alive during the examination of the tissue sections so that there would have been the possibility of applying a scan for cells within serum or primary cells. The evaluation of the tissue sections was not performed by an objectifiable measurement tool; but tissue sections were objectified by two independent investigators who evaluated the expression pattern blinded. There was no possibility to overview the process of the embedment of patient tissue, which leads to the fact that in the analyzed collective only tumorous tissue is accessible.

Double staining of FOXP3 and COX-2 in the immunofluorescence revealed evidence of nuclear expression of FOXP3, but also evidence of partially cytoplasmic expression of the transcription factor FOXP3. In the Human Protein Atlas, the reference images also show partial cytoplasmic expression, which is described as expression in the nucleoplasm [60].

The data of the collective do not contain information about the medication status of the patients at the time of tissue sampling; therefore, no conclusion can be drawn about a possible intake of a COX- inhibitor. However, it is unlikely that medication alters the expression pattern because drugs only affect activity and not expression. In addition, the serological test results of the patients are not available and subsequent blood diagnostics are not possible.

Comparable studies with such a high number of examined tissue sections from vulvar cancer patients do not currently exist. Only Sznurkowski et al. reported the results of their study of the influence of Treg cells in vulvar carcinoma; using a patient collective of 110 patients, they observed that Treg cells had no effect on overall survival [61].

In addition, a strong prognostic factor for survival of vulvar carcinoma patients is the nodal status [62]. In our study, Cox regression did not identify the nodal status as an independent prognostic factor. This can be explained by a high number of nodal-negative patients (42.6%), as well as an equally high number of unknown lymph node status (29.8%). The missing data mainly concerned patients whose date of diagnosis was decades ago. Because of this time gap between diagnosis and the analysis for this study, there were missing values in some cases.

Due to the rarity of this cancer, especially compared with other cancers such as breast cancer, the sample size of our collective in the current literature is a very large one. The increasing rate of new cases again highlights the greater relevance of the need for new knowledge regarding prognostic factors and linkage to tumor immunology in a world where the role of immunotherapy is rapidly gaining importance.

## 4. Materials and Methods

A total of 177 patients with VSCC primarily diagnosed in the period from 1990 to 2008 were included in this study. The entire patient group was treated at the department of Obstetrics and Gynecology of the Ludwig-Maximilians-University in Munich, Germany. Surgically obtained tissue samples were histopathologically processed and specified. All follow-up and survival data were provided by the Munich Cancer Registry (MCR) from the Munich Tumour Centre (TZM—Munich Tumour Centre, Munich, Germany). 

For immunohistochemical staining, 157 of the 177 samples were available. During the evaluation, a further 16 tissue samples were excluded, as the incisions did not contain a tumor, but only precancerous stages of the carcinoma. Therefore, in the end a collective of 141 slides of VSCC was assessed, one slide per staining and case.

The median age of the investigated collective was 70 years, ranging from 20 to 96 years, with 72 of the 141 patients younger than 70 years (=51.8% of the collective). All relevant clinic–pathologic parameters are listed in Table 3 below. The collective is the same as described by Ansorge et al. in previous studies by our research group [16].

### 4.1. Ethical Approval

All patients data were completely anonymized and the study was performed according to the standards set in the Declaration of Helsinki 1975. The examined tissues were residual material that had been collected in the first instance for histopathological diagnostic procedures. The actual study was approved in writing by the Ethics Committee of the Ludwig-Maximilians-University, Munich, Germany (approval number 367-16). The authors were blinded for clinical information during the experimental analysis.

### 4.2. Immunohistochemistry

The already formalin-fixed and paraffin-embedded samples were then cut by microtome to 4µm from the paraffin block and mounted on SuperFrost Plus microscope slides (Menzel Glaeser, Braunschweig, Germany). To deparaffinize the tissue, samples were processed with xylol for 20 min and washed by 100% ethanol. All slides were prepared with 3% hydrogen peroxide diluted in methanol for 20 min to stop activity of endogenous peroxidase. Afterwards, rehydration took place in a descending alcohol series (100%, 70%, 50%) and the samples were washed with distilled water. The samples were then heated with citric acid buffer in a pressure cooker to uncover antigen epitopes. Furthermore, slides were washed two times with phosphate buffered saline (PBS). A Zytochem-Plus HRP Polymer-kit (Zytomed, Berlin, Germany) was utilized for blocking and antibody staining. After saturating the electrostatic charges in the tissue with blocking solution for 5 min, either the polyclonal rabbit IgG anti- COX-2 antibody (Sigma, St. Louis, MI, USA, SAB4502491) or the polyclonal rabbit IgG anti-PPARγ antibody (abcam, Cambridge, United Kingdom, ab59256) was applied on tissue specimens. Anti-COX-2- antibody was diluted at a ratio of 1:400 and anti-PPARγ antibody at a ratio of 1:100. The incubation time of both antibodies amounted to 16 h at 4 °C in a humidity chamber. Slides were incubated by post-block reagent for 20 min and thereafter by HRP-Polymer for 30 min at room temperature in the humidity chamber. After each application with the antibody, post-block, and HRP-Polymer, the samples were washed two times with PBS. 3,3′-Diaminobenzidine (Dako, Hamburg, Germany) catalyzed the peroxidase substrate staining so that the color precipitation was detectable with a light microscope. Finally, the slides were counterstained with hemalum, washed again using 100% ethanol, and covered with glass. Both antibodies were stained in placenta tissue as a positive control to validate the staining method. The staining was considered positive in the case of cytoplasmic positivity of COX-2, and in the case of cytoplasmic and nuclear positivity of PPARγ. The semi quantitative immunoreactive score (IRS) by Remmele and Stegner [63] was used to evaluate immunostaining, together with a light microscope (Leitz, Wetzlar, Germany). For this purpose, a product of two factors, the intensity and the proportion of staining in the tumor tissue, was formed. The intensity was classified into 0 = no, 1 = weak, 2 = moderate, 3 = strong; the proportion of tumor tissue was also categorized: 0 = no staining, 1 ≤ 10%, 2 = 11% to 50%, 3 = 51% to 80%, 4 ≥ 81%. The antibodies showed expressions in cytoplasm and in nucleus, so both expression templates were examined independently by IRS. Patient data were correlated by IRS and by its two IRS-forming factors of staining intensity and percentage of positively stained cells. Neither HPV testing nor an immunohistochemical survey of p16 status was performed as part of this study; information on this was obtained from archives.

### 4.3. Immunofluorescence

The same formalin-fixied and paraffin-embedded samples were placed in xylol for 20 min for deparaffinization. Subsequently, the sections were panned in ethanol in order of descending concentrations (100%, 70%, 50%) and washed in distilled aqua. Unmasking of antigens was performed simultaneously with the immunohistochemistry protocol by a 5 min heat pretreatment in a pressure cooker in the citrate buffer-use solution described previously. After washing in distilled water and PBS for 4 min, incubation with UltraV block solution (Labvision, Fremont, CA, USA) was performed. The solution was tipped off after 15 min and the primary antibodies were applied. COX-2 was stained at a ratio of 1:400 together with CD56 (Bio-Rad, Oxford, United Kingdom, MCA591) at a ratio of 1:100, CD68 (Sigma, St. Louis, MO, USA, AMAb90874) at a ratio of 1:8000, or FOXP3 (Abcam, Cambridge, United Kingdom, ab20034) at a ratio of 1:300 in dilution medium (Dako, Glostrup, Denmark, S3022) to differentiate cells in a double staining procedure. CD56 is a known structural protein of natural killer (NK) cells [64], CD68 is known as a structural protein of macrophages, and FOXP3 is expressed specifically by regulatory T (Treg) cells [65,66,67]. Double staining with PPARγ and CD68 was performed to subtype macrophages [68]. Here, a concentration of 1:100 was chosen for the PPARγ antibody (Abcam, Cambridge, United Kingdom, ab27649) and CD68 was added, as in the double staining with COX-2.

Incubation was performed for 16 h at 4 °C. After washing in PBS, the experimental room was darkened and the mixed secondary antibodies were applied: Cy-2- conjugated antibody at a ratio of 1:100, which later fluoresced green (Dianova, Hamburg, Germany, 115-546-062) or Cy-3- conjugated antibody at a ratio of 1:500, which fluoresced red (Dianova, Hamburg, Germany, 111-225-144). After 30 min of incubation, the excess secondary antibodies were washed off in PBS. In the dark, the preparations dried at room temperature and were cover slipped with Mounting medium for fluorescence with DAPI, which stains the cell nuclei as a blue light impression.

During the performance of immunofluorescence double staining, the primary antibodies CD56, CD68 and FOXP3 appear green in the fluorescence microscope and COX-2 is perceived as red fluorescence. The double staining was evaluated and assessed using a fluorescence microscope (Zeiss, Oberkochen, Germany). A total of 18 tumor microenvironments were examined in a sampling from the previously described patient population using this described method, and based on this, subtyping of stromal tumor-infiltrating lymphocytes was performed.

### 4.4. Statistical Analysis

For statistical analysis, SPSS Statistic version 25 (IBM Corp., Armonk, NY, USA) was used. The non-parametric Kruskal–Wallis test was used to compare between and among groups. Correlation analyses were performed using the Spearman rank correlation coefficient. Kaplan–Meier curves were generated from the collected survival data of patients with vulvar carcinoma. The differences between these curves of sTILs with high and with low COX-2 expression were tested with the log-rank test. The identification of these sTILs was analyzed by specific markers of Treg cells, macrophages and NK cells. Cox regression models were applied for multivariate analysis. Patient-specific pairwise analysis of significance differences in immune cell subtypes of Treg cells, NK cells, and macrophages was performed using the Wilcoxon test. The level of statistical significance was accepted at *p* ≤ 0.05 and all tests were two-sided. 

## 5. Conclusions

In this study, COX-2 positive sTILs were observed to be an independent prognostic factor in long-term overall survival in the patient population studied. Furthermore, subtyping by immunofluorescence revealed that most sTILs are Treg cells. This is the first time that COX-2 positive Treg cells have been observed to have a significant impact in the context of long-term survival in patients with vulvar cancer. This finding might help on the way towards individualized immunotherapy and could eventually lead to further investigations of new immune checkpoints. Further research goals are in vitro experiments to reliably demonstrate a direct causal relationship between COX-2 and Treg cells and the planning of a prospective study model using COX-2 inhibition to add an important further approach to the limited therapy options and prognosis of vulvar cancer.

## Figures and Tables

**Figure 1 ijms-23-04662-f001:**
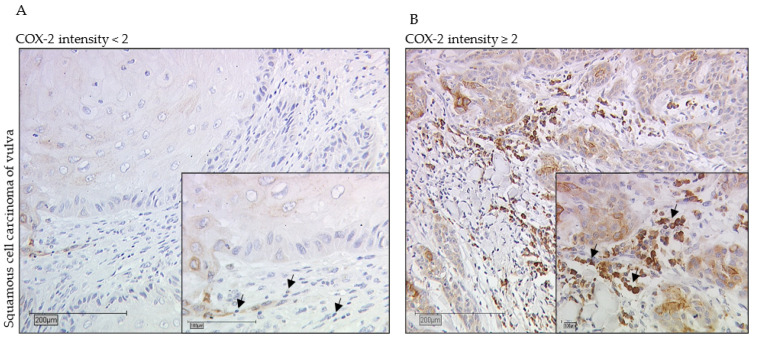
Immunohistochemistry staining of COX-2 (10× and 25× magnification) in vulvar cancer tissue showing expression of stromal sTILs with intensity of COX-2 expression < 2 (**A**) and ≥2 (**B**). The arrows in both figures mark exemplarily sTILs (**A**,**B**). The Kaplan–Meier curve shows a significantly worse overall survival rate for the patients with a strong COX-2 intensity of sTILs > 2 in long- term survival of 10 years (* *p* = 0.013, (**C**)). (**C**) is labeled with the percentage of patients with vulvar carcinoma still alive after 10 years: In the group of patients with COX-2 expression of sTILs ≤ 2 of vulvar carcinoma, 79% are still alive, whereas among the diseased women with higher COX-2 expression of sTILs, only 52% are still alive in comparison.

**Figure 2 ijms-23-04662-f002:**
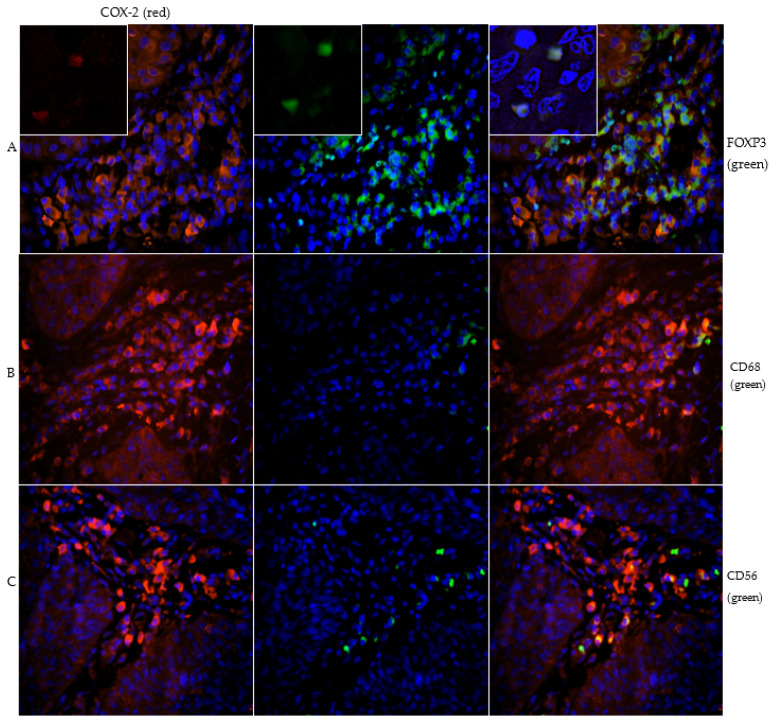
Image series (**A**) demonstrates the double staining of COX-2 (red) and FOXP3 (green) with inserts of magnification 40× to show nuclear staining of FOXP3. The majority of COX-2 positive immune cells peritumorally are FOXP3 positive and thus detected as Treg cells. Image series (**B**) presents the double staining of COX-2 (red) and CD68 (green). The subtyping shows that some macrophages are COX-2 positive. Figure series (**C**) shows the double staining of COX-2 (red) and CD56 (green). The CD56 positive NK cells are weakly pronounced and only singly distributed in the stroma.

**Figure 3 ijms-23-04662-f003:**
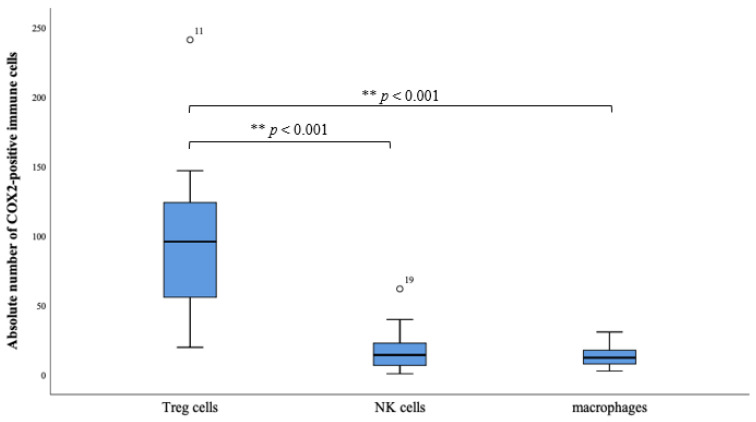
The boxplots reveal the absolute number of COX-2 positive Treg cells (FOXP3+), NK cells (CD56+) and macrophages (CD68+). There is a clear distribution in the direction of the Treg cells. The box plots indicate mild outliers, which are marked with circles. These outliers (case numbers 11 and 19) show an interquartile distance to the third quartile of values that is less than three times higher than the third quartile of values. The difference of the occurrence of the subtypes Treg cells to NK cells, as well as Treg cells to macrophages, is highly significant (** *p* < 0.001).

**Figure 4 ijms-23-04662-f004:**
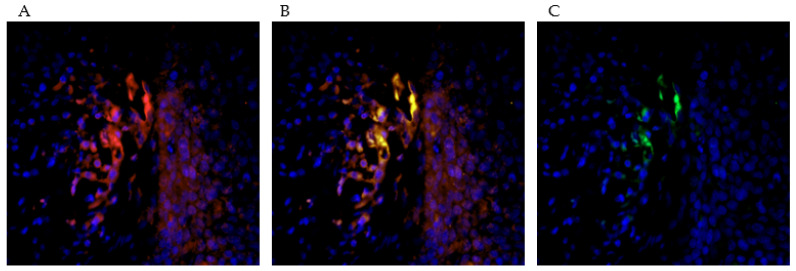
The picture series (**A**–**C**) demonstrate the double staining with PPARγ (red) and CD68 (green). This allows a subtyping of CD68-positive macrophages into M2-polarized and M2-unpolarized macrophages. PPARγ acts as a marker for the M2-polarization of the macrophages. In the selected tissue sections, it was found that more doubly stained, M2-polarized macrophages are resident in the immediate environment and within the tumor association than in the extratumoral stroma.

**Figure 5 ijms-23-04662-f005:**
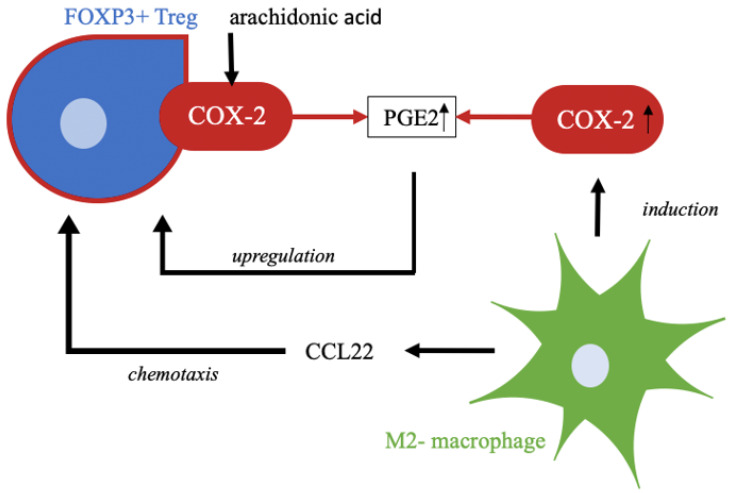
COX-2 produces PGE2, a prostaglandin, from the substrate arachidonic acid as part of lipid metabolism. Upregulation of Treg cells is affected by this prostaglandin. M2-polarized macrophages attract regulatory T cells by chemotaxis, e.g., by CCL22. In addition, M2-polarzied macrophages induce expression of COX-2, which in turn leads to higher enzyme activity and thus higher accumulation of products, such as PGE2 [27,28,48,54,55].

**Table 1 ijms-23-04662-t001:** There is a clear difference in long-term overall survival after 10 years for patients with intensity values for COX-2 expression in sTILs > 2. Patients with intensity values > 2 live with a median of 129 months, whereas patients with lower IRS values survive 216 months. As the table shows, the survival of patients of both groups differs by 87 months, i.e., more than 7 years. Even in the total group, a survival difference is recorded: patients live a total of 163 months, but still lose 34 months of life with higher intensity values.

Median for Long-Term Overall Survival Time (Months) after 10 Years			
COX-2 Intensity in sTILs	Estimate	Lower 95% CI	Upper 95% CI
IRS ≤ 2	216.000	37.953	290.388
IRS > 2	129.000	14.736	157.882
Overall	163.000	21.286	204.720

**Table 2 ijms-23-04662-t002:** Cox regression of clinical–pathological variables regarding long-term overall survival after 10 years in VSCC patients.

Variable	Significance	Hazard Ratio of Exp (B)	Lower 95% CI of Exp (B)	Upper 95% CI of Exp (B)
COX-2 intensity in sTILs	0.007	4.731	1.525	14.676
pT	0.063	6.576	0.192	2.463
pN	0.633	0.996	0.980	1.012
Grading	0.078	2.204	0.914	5.312
FIGO	0.565	0.687	0.192	2.463
p16 status	0.024	0.243	0.071	0.831

**Table 3 ijms-23-04662-t003:** Clinicopathological parameters of VSCC patients’ collective.

Clinicopathologic Parameters	*n*	Percentage (%)
**Histology**		
keratinizing	134	95
warty/basaloid	7	5
**Tumor size**		
T1	51	36.2
T2	74	52.5
T3	9	6.4
unknown	7	5
**Nodal status**		
N0	60	42.6
N1	31	22
N2	8	5.7
unknown	42	29.8
**FIGO**		
I	45	31.9
II	45	31.9
III	36	25.5
IV	9	6.4
unknown	6	4.3
**Grading**		
G1	24	17
G2	87	61.7
G3	29	20.6
NOS/unknown	1	0.7
**p16 status**		
positive	34	24.1
negative	57	40.4
unknown	50	35.5
**COX-2 expression of sTILs**		
Positive	136	96.5
negative	4	2.8
unknown	1	0.7
**Progression status**		
positive	61	43.3
negative	79	56
unknown	1	0.7
**Local recurrence status**		
positive	35	24.8
negative	105	74.5
unknown	1	0.7

## Data Availability

The data presented in this study are available on request from the corresponding author. The data are not publicly available due to ethical issues.

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
