# Peer review of "Regulatory T Cells with Additional COX-2 Expression Are Independent Negative Prognosticators for Vulvar Cancer Patients"

_ijms, 2022, doi:10.3390/ijms23094662_

Round 1

Reviewer 1 Report

Authors have attempted to look at COX2 and FoxP3 expression in tumor infiltrating cells from patients with vulvlar cancer. The research question is highly important and there is need for identification of tools for prognosis. However, the manuscript lacks originality. The authors have used the same cohort and same marker for staining tissues as in their last manuscript on COX-2 in vulvar cancer (apparently the same information as in the last manuscript Ansorge et al. Diagnostics 2021). Authors mention that COX2 is an independent prognostic factor in vulvar cancer in their last manuscript. reader finds it difficult to understand the context of using the same cohort and same stainings. since methods and cohort are exactly same, authors must only cite their previous work and must explain what new information this manuscript is adding as compared to their last manuscript. How much difference in prognosis it adds upon looking at COX2 and foxP3.

Authors have attempted to do immunofluorescence staining on sections, but the image resolution and magnification is too less to make any inferences. Authors used CD56 staining to show that FoxP3+ cells are expressing COX-2 but not CD56+ cells. However, they didn't use CD3 antibody to make sure that FoxP3+ cells are indeed T cells. Red channel seems overexposed. Authors must include staining controls in order to show the specificity and perform 4 color immunofluorescence at high resolution.  

Here are some more comments: 

  1. Cox2 can be both pro and anti-inflammatory depending on which polyunsaturated fatty acids, it makes. how do authors perceive the role of this molecule in context of the cancer? This can be an addition to the story the authors are trying to make.
  2. Did authors find combination of foxP3 and CoX2 to be better prognostic marker than the combination used in their previous manuscript?

Author Response

Reviewer Responses

Reviewer 1:

First of all, we would like to thank you for reviewing the paper  and your interesting comments and remarks. In the following you will find our answers to your questions along with descriptions of what we reworked or added to our updated version of the paper.

  1. Authors mention that COX2 is an independent prognostic factor in vulvar cancer in their last manuscript. reader finds it difficult to understand the context of using the same cohort and same stainings. since methods and cohort are exactly same, authors must only cite their previous work and must explain what new information this manuscript is adding as compared to their last manuscript.

A remark has been added that the patient collective studied is the same collective of the previous publication.

“The collective is the same as described by Ansorge et al. in previous studies of our research group. [16]“ (line  267-268 in manuscript)

Regarding the methods, the current and past publication differ from each other. Though the method of immunohistochemistry was used again, in this study the staining behavior of the stromal surrounding cells was evaluated and thus not, as in the previously published work of Ansorge et al., the vulvar carcinoma cells. In addition, immunofluorescence was performed as additional method for specific subtyping of stromal immune cells. In general the first paper was about cancer cell expression of COX-2 and PPARg, whereas in this paper we characterized specific COX-2 and PPARg positive immune cells.

  1. How much difference in prognosis it adds upon looking at COX2 and foxP3.

Did authors find combination of foxP3 and CoX2 to be better prognostic marker than the combination used in their previous manuscript?

In our previous paper, the expression of COX-2 in vulvar cancer cells was studied as a prognostic factor for overall survival; however, in this paper, the expression of COX-2 in specific immune cells in the direct environment of vulvar cancer tissue, i.e., in the surrounding stroma, is studied in long-term survival. Here, we have found that these cells, which are not located in the tumor tissue itself but in the surrounding stroma, are FOXP3 positive immune cells, specifically Treg cells. This finding is new and not investigated before.

Thus, immunological processes have an impact on the prognosis of affected patients and may be a potential target for systemic therapy.

“Immunological processes in the context of tumorgenesis and maintenance of tumor growth rely on enhancing effect as tumor promotion and on inhibitory effects as tumor suppression. For M2-polarized macrophages and Treg cells in the tumor environment a tumor promoting effect is reported. [48]

Other studies observed that population of Treg cells in sTILs is significantly higher in carcinoma tissue than in normal healthy tissue [49,50]. Our study showed a clear majority of FOXP3 positive Treg cells with 70.2% (Figure 2, 3) within the sTILs, so a negative impact on long term survival in patients with VSCC is assumed. A significant impact of immunologic processes on long term survival is already known and has been described before [51], but for ovarian cancer.” (line 186-195 in manuscript)

  1. Authors have attempted to do immunofluorescence staining on sections, but the image resolution and magnification is too less to make any inferences.

We have checked, changed and optimized the image series of Figure 2 completely in the illustration and enlargement due to the comment. We found a mistake in labeling the image series, so the actual version is totally revised und correct. (Figure 2 in manuscript)

  1. Authors used CD56 staining to show that FoxP3+ cells are expressing COX-2 but not CD56+ cells.

Image series C of Figure 2 shows the double immunofluorescence staining of the markers CD56 and COX-2. CD56 appears green, whereas COX-2 appears red in the plot. This staining does not show that FOXP3 positive cells express COX-2, but that the COX-2 positive sTILs contain only a small number of CD56 positive cells. CD56 is an established marker for subtyping immune cells to detect natural killer (NK) cells.

…“Figure series C shows the double staining of COX-2 (red) and CD56 (green). The CD56 positive NK cells are weakly pronounced and only singly distributed in the stroma.”
(Figure 2 capture in manuscript)

  1. However, they didn't use CD3 antibody to make sure that FoxP3+ cells are indeed T cells.

We used the more specific and established marker FOXP3 for subtyping of regulatory T cells (Treg cells). We added some more references to the specifity of FOXP3 marker for subtyping Treg cells.

CD56 is a known structural protein of natural killer (NK) cells [63], CD68 is known as a structural protein of macrophages, and FOXP3 is expressed very specifically by regulatory T (Treg) cells [64-66].“ (line 334-336 in manuscript)

  1. Red channel seems overexposed.

We adjusted the intensity of the red channel again and thank you for this hint.
(Figure 2A-C in manuscript)

  1. Authors must include staining controls in order to show the specificity and perform 4 color immunofluorescence at high resolution.  

We added staining controls (immunohistochemistry and immunofluorescence) in supplementary material (Supplementary Figure S4 and S5).

  1. Cox2 can be both pro and anti-inflammatory depending on which polyunsaturated fatty acids, it makes. how do authors perceive the role of this molecule in context of the cancer? This can be an addition to the story the authors are trying to make.

We briefly described the role of COX-2 in the context of carcinogenesis in the introduction.

“It is thought that products of COX-2 such as prostaglandin E2 (PGE2) influence critically the development of tumors e.g., in angiogenesis [28, 29].” (line 83-84 in manuscript)

Thanks to your suggestions, we have expanded this small part and described it in more detail in the introduction part, as well as supporting it with additional references.

Effecting COX-2 products, the prostanoids, appear to be associated with the development and progression of tumor disease. Factors such as angiogenesis, invasion, apoptosis inhibition, growth and aggressiveness of the tumor seem to be highly dependent on COX-2 and its products.[28, 29].” (line 80-83 in manuscript)

Reviewer 2 Report

The study finds that COX expression in TReg cells is predictive of survival for women with vulvar cancer.  However, the description of findings must be clarified.  In the abbreviation sTIL, what does the small "s" mean (stromal ?). Yet in the title and in the details of the report, the focus is on TReg cells.   The Kaplan Meier plots in Figures 1C, 1D and 1E must be clarified before publication.  Figure 1C does not show a significant difference in survival with high COX2 (P<0.20) whereas Figures 1D and 1E show a significant difference (P<0.01) with high COX2. Figures 1D and 1E actually show the same K-M plot and should be combined.  Usually, the medians of each group are shown for comparison with a P value.  In particular, what cells are being stained in Figure 1C (no signficant difference in survival with high COX2) versus Figures 1D and 1E (significant difference in survival with high COX2).  Clarification of these figures is critical to the reader being able to understand the results.  Do Figures 1D and 1E refer only to COX2 expression in TReg cells whereas Figure 1C refers to total COX2 in sTIL?  If so, this should be clearly stated in the results and in the Figure caption.  If the significant results in survival pertain only to the COX2 expression in TReg cells rather than sTIL, then the abstract and text needs to be revised to reflect this specificity.   Furthermore, this should be stated in the methods section, e.g.,  K-M plots, log rank tests and Cox regression analysis were used to examine survival in patients with high versus low COX2 expression in sTIL and TRegs.   

Author Response

Reviewer 2:

First of all, we would like to thank you for reviewing the paper and your interesting comments and remarks. In the following you will find our answers to your questions along with descriptions of what we reworked or added to our updated version of the paper.

  1. However, the description of findings must be clarified.  In the abbreviation sTIL, what does the small "s" mean (stromal ?). Yet in the title and in the details of the report, the focus is on TReg cells.   

The “s” in sTIL means stromal. We added – beside the list of abbreviation- a detailed explanation of the term “sTILs” in the part of introduction:

„We used the abbreviation sTILs (stromal tumor-infiltrating lymphocytes) in our study for these cells. They have become important players in immuno-oncology with regard to predicting prognosis in cancer patients. [34-37]“ (line 85-88 in manuscript)

...

“Types of sTILs and iTILs form specific group of immune cells called tumor infiltrating lymphocytes (TILs) and were assessed based on the recommendations of the International TIL Working Group (ITILWG). According to them, lymphocytes organized in tumor nests are defined as intratumoral TILs (iTILs). They are making cell-to-cell contact without intervening stroma and interacting directly with carcinoma cells. Hence, stromal TILs (sTILs) are located in the stroma among carcinoma cells and having no direct contacts with carcinoma cells.[40] In this study, we investigated especially sTILs and no iTILs.
This study analyzed and characterized sTILs expressing COX-2 as mainly Treg cells in tissues of VSCC, their relevance as prognostic factor, and in addition subtyping and polarization of infiltrating macrophages in tumor microenvironment.” (line 93-102 in manuscript)

In the title, the focus is on Treg cells, because we succeeded in specifying the immunohistochemically COX-2 positive sTILs in more detail on the basis of immunofluorescence and identifying them as Treg cells, so that the title already contains one of the main results. We explain this in the abstract and revised it again:

“The aim of this study was to analyze the influx of COX-2 positive tumor infiltrating lymphocytes and monocytes and its influence on prognosis. Using subtyping by immunofluorescence, the majority of COX-2 expressing immune cells were identified as FOXP3-positive regulatory T cells.“ (line 19-22 in manuscript)

  1. The Kaplan Meier plots in Figures 1C, 1D and 1E must be clarified before publication.  Figure 1C does not show a significant difference in survival with high COX2 (P<0.20) whereas Figures 1D and 1E show a significant difference (P<0.01) with high COX2. Figures 1D and 1E actually show the same K-M plot and should be combined. 

In Figure 1C, we intentionally wanted to visualize the lack of significance of COX-2 expression of sTILs in patients with vulvar cancer in overall survival to clarify the contrast in the following Figure 1D. Because of your assumption we moved Figure 1C in the part of supplementary (now Supplementary Figure S3)

Indeed, Figure 1D shows a strong significance in long-term survival at 10 years.

The recommendation of the reviewer to optimize the previously chosen representation of Figure 1D and Figure 1E was gratefully acknowledged and we reduced them to one Kaplan-Meier curve, so that there is now only Figure 1D, changed into Figure 1C. The figure caption was adapted accordingly.

  1. Usually, the medians of each group are shown for comparison with a P value. 

We showed this data represented in Table 1.

  1. In particular, what cells are being stained in Figure 1C (no signficant difference in survival with high COX2) versus Figures 1D and 1E (significant difference in survival with high COX2).  Clarification of these figures is critical to the reader being able to understand the results. 
    Do Figures 1D and 1E refer only to COX2 expression in TReg cells whereas Figure 1C refers to total COX2 in sTIL?  If so, this should be clearly stated in the results and in the Figure caption. 

We agree in reviewer’s assumption and move the Figure 1C in the supplementary (now added in Supplementary Figure S3). We think we should keep this figure in the supplementary at least, because there is a clear trend of disadvantage in the overall survival.

In Figure 1C , 1D and 1E (1D and 1E now combined in Figure 1C) it is shown that the intensity of COX-2 expression of the sTILs of patients with vulvar carcinoma has a significant effect on long-term survival only after 10 years. The difference in significance between the Kaplan-Meier curves shown in Figure 1C, 1D, and 1E (1D and 1E now combined in Figure 1C) is due to the fact that Figure 1C looks at overall survival and Figure 1D looks at long-term survival at 10 years.

“The Kaplan-Meier curve shows a significantly worse overall survival for the patients with strong COX-2 intensity of sTILs > 2 only in long- term survival of 10 years (*p= 0.013, Figure 1C). In Figure 1C, it is labeled by the percentage of patients with vulvar carcinoma still alive after 10 years: In the group of patients with COX-2 expression of sTILs ≤ 2 of vulvar carcinoma, 79% are still alive, whereas among the diseased women with higher COX-2 expression of sTILs, on the other hand, only 52% are still alive in comparison.”
(corrected part of caption of Figure 1A-1C in manuscript)

COX-2 positive sTILs are predominantly FOXP3+ cells, as demonstrated in the subtyping of sTILs, so that assignment to the regulatory T cell subtype was done and the conclusion is stated, that the seen effect of COX-2 expression of sTILs on long-term survival after classification by immunofluorescence is attributed to regulatory T cells and therefore the choice of the title of the publication "Regulatory T cells with additional COX-2 expression are independent negative prognosticators for vulvar cancer patients" remains.

  1. If the significant results in survival pertain only to the COX2 expression in TReg cells rather than sTIL, then the abstract and text needs to be revised to reflect this specificity.   

We changed the abstract as suggested by the reviewer.

As this review already stated, subtyping of sTILs was performed: Here, the markers CD68, CD56, and FOXP3 were used to show which specific subgroups the sTILs studied are. Among the sTILs, a clear majority of FOXP3 positive cells was found, leading to the conclusion that regulatory T cells are a negative prognosticator for patients with vulvar carcinoma in long-term survival.

“Vulvar cancer incidence numbers have been steadily rising over the past decades. Especially the number of young patients with vulvar cancer increased recently. Therefore, the need to identify new prognostic factors and in addition, therapeutic options for vulvar carcinoma is more apparent. The aim of this study was to analyze the influx of COX-2 positive tumor infiltrating lymphocytes and monocytes and its influence on prognosis. Using subtyping by immunofluorescence, the majority of COX-2 expressing immune cells were identified as FOXP3-positive regulatory T cells. In addition, peri- and intra-tumoral macrophages in the same tumor tissue were detected simultaneously as M2-polarized macrophages. COX-2 positive immune cells were independent negative prognostic markers in long-term overall survival of patients with vulvar cancer. These results show an influence of immune cell infiltration for vulvar carcinoma patients. Immune cell infiltration and immune checkpoint expression may therefore become interesting targets for further research on new vulvar cancer treatment strategies.”
(lines 16-27 in manuscript)

  1. Furthermore, this should be stated in the methods section, e.g.,  K-M plots, log rank tests and Cox regression analysis were used to examine survival in patients with high versus low COX2 expression in sTIL and TRegs.   

We modified the methods section as suggested to more clearly link the statistical tests to the results and thereby explain them in more detail:

“Kaplan-Meier curves were generated from the collected survival data of patients with vulvar carcinoma. The differences between these curves of sTILs with high and with low COX-2 expression were tested with the log-rank test. The identification of these sTILs was analyzed by specific markers of Treg cells, macrophages and NK-cells.” (line 361-364 in manuscript)

“Patient-specific pairwise analysis of significance differences in immune cell subtypes of Treg cells, NK- cells, and macrophages was performed using the Wilcoxon test.” (365-367 in manuscript)

Round 2

Reviewer 1 Report

Authors have made an attempt to revise the manuscript. However, as previously mentioned, immunofluorescence is central to this manuscript and it is more appropriate to consider this as a short letter in view of their previous manuscript. Immunofluorescence staining of FoxP3 looks really on the surface of the cells, whereas being a transcription factor, it should be colocalised with the nucleus and thereby appear as merged with DAPI. In the representative figure, it appears as non-specific staining. Authors must show 40X and 63X images in order to build confidence on the staining and thereby their interpretations. It would also make sense to show some more stainings as supplementary stainings. In the control stainings, authors have 2 black figures and mentioned as Isotypes for 5 different markers. However, this should have been isotype for one marker in one figure while including other real antibodies. In my opinion, it is very important to have properly resolved figures at appropriate magnification and showing nuclear FoXP3 staining.

Author Response

Reviewer Response

Reviewer 1:

First of all, we would like to thank you for reviewing the paper  and your interesting comments and remarks. In the following you will find our answers to your questions along with descriptions of what we reworked or added to our updated version of the paper.

  1. Authors have made an attempt to revise the manuscript. However, as previously mentioned, immunofluorescence is central to this manuscript and it is more appropriate to consider this as a short letter in view of their previous manuscript.

Immunofluorescence staining of FoxP3 looks really on the surface of the cells, whereas being a transcription factor, it should be colocalised with the nucleus and thereby appear as merged with DAPI. In the representative figure, it appears as non-specific staining. Authors must show 40X and 63X images in order to build confidence on the staining and thereby their interpretations.

Changes made: In the representative images in the manuscript, the staining of FOXP3 appears superficial and blurred with DAPI. Therefore, staining was performed again to support the statement of the manuscript and to show that FOXP3 is stained as a transcription factor in the nucleus. Furthermore, it was shown that in some cases cytoplasmic staining of FOXP3 also occurs, but this is also described in the Human Protein Atlas and added as a reference.

The additional staining of FOXP3 at the desired magnification (40x) was included in the manuscript and also explained in part of discussion.

In summary, we reevaluated 30 cases of our patient collective showing COX-2/FOXP3- double staining.

Figure 2. Image series A demonstrates the double staining of COX-2 (red) and FOXP3 (green) with inserts of magnification 40x to show nuclear staining of FOXP3. …“

“Double staining of FOXP3 and COX-2 in the immunofluorescence revealed evidence of nuclear expression of FOXP3, but also evidence of partially cytoplasmic expression of the transcription factor FOXP3. In the Human Protein Atlas, the reference images also show partial cytoplasmic expression, which is described as expression in the nucleoplasm.[60] (line 232-236 in manuscript)”

  1. It would also make sense to show some more stainings as supplementary stainings. In the control stainings, authors have 2 black figures and mentioned as Isotypes for 5 different markers. However, this should have been isotype for one marker in one figure while including other real antibodies. In my opinion, it is very important to have properly resolved figures at appropriate magnification and showing nuclear FoXP3 staining.

Changes made: It was commented that further supplementaries should be added to support the staining methods. The negative controls are based on the control of the secondary antibodies used and have been included in Figure S5 and Figure S6 as inserts (Figure S5c and Figure S6c). The black images are generated by testing the Cy2/3-binding secondary antibodies. COX-2 is a rabbit antibody visualized by goat-anti-rabbit Cy-3, whereas FOXP3, CD56 and CD68 are mouse antibodies visualized by goat-anti-mouse Cy-2. A black image appears in the negative control because the secondary antibody did not produce unspecific staining without the corresponding primary antibody.

The additional staining of FOXP3 at the desired magnification (40x) was included in the manuscript (Figure 2 in manuscript).

Reviewer 2 Report

The revisions appear sufficient.

Author Response

Thank you very much for your evaluation.

Round 3

Reviewer 1 Report

I don’t see any improvement in the staining figures for FoxP3, which is the sole new finding in this paper as compared to their previous publications. I don’t find merit in further review without required changes and would rather choose to not review it further. 

This manuscript is a resubmission of an earlier submission. The following is a list of the peer review reports and author responses from that submission.

Round 1

Reviewer 1 Report

The authors describe the Cox2 expression in tumor associate leucocytes in a cohorte of 177 vulvar cancers. They found a negative correlation of cox 2 expression in Tils to overall survival of vulvar cancer patients. Furthermore, an association of TILs to specialized macrophages was found. The results are discussed in the light of the present literature in this field.

There are some issues in this paper.

  1. The authors did not describe the percentage of vulvar carcinomas with and without TILs in their material. The intensity of TILs should be quantified.
  2. I could not found the number of cases with TILs with COX2 expression and without cox2 expression.
  3. There are 3 survival curves with in this paper. What is the reason? 1 should be enough.
  4. Cox2 expression in Tils was the only independent parameter. Nodal status is a known important prognostic parameter in vulvar cancer. However, it was not significant in this patient cohort. The authors should discuss this point.
  5. I can’t detect Tils in Figure 1A. Therefore, the negative statement is questionable. A different picture should be provided.
  6. In line 36 the authors write un-keratinized. Non–keratinized is mor in use. In the introduction there is a lot about HPV vaccination in vulvar cancer without reasonable relation to the analysis. This is not necessary.

Reviewer 2 Report

Abstract:

Lines 20-21: ­i would suggest to use “were” instead of “are”, in concordance with how the entire manuscript is written. In addition, this phrase ideally should go after the methods have been described.

Lines 36-42: include a couple of sentences on the two pathways leading to VSCC: one HPV-associated and another HPV-independent. One useful citation reference: “Del Pino M, Rodriguez-Carunchio L, Ordi J. Pathways of vulvar intraepithelial neoplasia and squamous cell carcinoma. Histopathology. 2013 Jan;62(1):161-75”. State also that HPV-negative VSCC affecting older women are predominant in high-income countries, in contrast with HPV-associated VSCC affecting younger women are prevailing in low-income settings.

Line 39: replace by “lichen sclerosus”.

Line 43-44: I would add “radical“ and replace “manifestations” by “involvement”.

Line 52: add “aging population” as a contributing factor as well.

Lines 56-58: “additional threat” to what? authors should also think about that HPV-associated VSCCs are more prevalent throughout EU countries. Thus, low vaccination perhaps is not the risk factor which increases the most the prevalence of VSCC in Germany.

Line 69: replace “leucocytes” by “lymphocytes (also in the line 76). Also, once the sTILs term has been introduced, it has to be used throughout the manuscript.

Line 70: add some references to support the information on the role of TILs.

Line 76: specify which type of vulvar carcinomas: squamous (VSCC)? The authors introduced the term VSCC but then it is rarely used throughout the manuscript.

Materials and methods:

Entire section:

  • Was an HPV testing performed?
  • Was an histological revision conducted? (I suppose p16 results were also retrieved from the archives)

Line 233: specify if all 177 cases were squamous cell carcinomas?

Line 244: correct “= 51.8%%”

Line 242: specify how many slides per case were evaluated

Table 3:

  • Ideally should go to a Results section
  • Grading in VSCC has proven to not be useful, so I would not include it here

Lines 255: if the samples retrieved were from 1990-2008, then all these samples are already formalin-fixed and paraffin-embedded. Please, correct. Also, the FFPE samples have to be cutted with microtome into slides-add this as well.

Lines 257: “tissue pattern”. Perhaps authors referred to “tissue samples”?

Lines 279: to evaluate “immunostaining” and not “tissue patterns”

Lines 278-286: specify for each antibody if staining was considered positive in case of nuclear or cytoplasmic positivity

Line 288: “paraffin-embedded” instead of “paraffin-embedding”

Results:

Lines 82: this furst sentence shoulg go to M&M, and here, in Result section, it has to be stated that Figure 1A and 1B show..

Reviewer 3 Report

This study reported a novel observation that COX-2 positive Treg cells have a significant impact in the context of long-term survival in patients with vulvar cancer. However, the author have not proven the causal relationship which weaken the significance.

It will be better if the author have some results that indicate the intervention of COX-2 could stubborn the progression of vulvar cancer. 

There's several spelling mistakes in the paper, such as Ethical approval in line247.